# Effects of Three Different Family-Based Interventions in Overweight and Obese Children: The “4 Your Family” Randomized Controlled Trial

**DOI:** 10.3390/nu13020341

**Published:** 2021-01-24

**Authors:** Panagiotis Varagiannis, Emmanuella Magriplis, Grigoris Risvas, Katerina Vamvouka, Adamantia Nisianaki, Anna Papageorgiou, Panagiota Pervanidou, George P. Chrousos, Antonis Zampelas

**Affiliations:** 1Department of Food Science and Human Nutrition, Agricultural University of Athens, Iera odos 75, 118 55 Athens, Greece; pvaragiannis@aua.gr (P.V.); emagriplis@aua.gr (E.M.); grisvas@aua.gr (G.R.); kvamvouka@aua.gr (K.V.); anisianaki@aua.gr (A.N.); apapageorgiou@aua.gr (A.P.); 21st Department of Pediatrics, Medical School, National and Kapodistrian University of Athens, Mikras Asias 75, 115 27 Athens, Greece; ppervanidou@med.uoa.gr (P.P.); chrousos@gmail.com (G.P.C.); 3University Research Institute of Maternal and Child Health and Precision Medicine and UNESCO Chair on Adolescent Health Care, Medical School, National and Kapodistrian University of Athens, Aghia Sophia Children’s Hospital, 115 27 Athens, Greece

**Keywords:** childhood obesity, nutrition education, dietary intervention

## Abstract

Childhood overweight and obesity prevalence has risen dramatically in the past decades, and family-based interventions may be an effective method to improve children’s eating behaviors. This study aimed to evaluate the effectiveness of three different family-based interventions: group-based, individual-based, or by website approach. Parents and school aged overweight or obese children, 8–12 years of age, were eligible for the study. A total of 115 children were randomly allocated in one of the three interventions, and 91 completed the study (79% compliance); Group 1 (*n* = 36) received group-based interventions by various experts; Group 2 (*n* = 30) had interpersonal family meetings with a dietitian; and Group 3 (*n* = 25) received training through a specifically developed website. Anthropometric, dietary, physical activity, and screen time outcomes were measured at baseline and at the end of the study. Within-group comparisons indicated significant improvement in body weight, body mass index (BMI)-z-score, physical activity, and screen time from baseline in all three study groups (*p* < 0.05). Furthermore, total body fat percentage (%TBF) was also decreased in Groups 2 and 3. Between-group differences varied with body weight and %TBF change, being larger in Group 3 compared to Groups 1 and 2, in contrast to BMI-z-score, screen time, and health behaviors, which were significantly larger in Group 2 than the other two groups. In conclusion, personalized family-based interventions are recommended to successfully improve children’s lifestyle and body weight status.

## 1. Introduction

Globally, the prevalence of overweight and obesity among children is high and is considered a major public health issue. Children’s obesity prevalence is termed “an epidemic in the 21st century”, despite the plateau observed in some high-income countries [1]. In Greece, the prevalence of obesity in children aged 7–12 rose from 3.6% in 1992 [2] to 11.8% in 2007 [3] and 10.6–12.9% in 2013 [4]. Although these studies measured different children cohorts and are not nationally representative, an increasing trend, despite the already high prevalence, can be hypothesized. This is a public health problem, since the risk of developing chronic non-communicable diseases (NCDs) in adulthood is higher in overweight and obese children [5], and therefore, it is of utmost importance to identify effective interventions for this population. The main challenge is how best to intervene in order to accomplish body weight reduction in children.

Expert panels have highlighted the importance of focusing efforts on the reduction of excessive body weight in children at a relatively young age and including the family in this process [6,7]. Observational studies and broad family-focused intervention studies emphasize the impact of parenting on childhood overweight and obesity [8]. These family-based interventions have been proposed to combat pediatric obesity, since most interventions that focus on children only seem to result in null or non-clinically significant effects [7]. For children under 12 years of age, there is evidence that family-based programs that include behavioral treatment, physical activity, and dietary change are effective in reducing overweight/obesity [5,8,9]. Overweight and obesity status at a young age can lead to excessive body weight gain in the future [10,11], and children have a relatively short history of unhealthy habits, potentially making it easier to alter these behaviors than in adult populations [10,11]. Therefore, the earlier the intervention, the more probable the achievement of the change aimed for, while intervening at the family level may provide greater change and longer sustainability [6,7]. Parent involvement is an important component of child obesity prevention; however, the most effective way remains unclear. In addition, for the same reasons, children have better chances of increasing the duration and the level of different types of physical activity compared to adults [12].

Physical activity level and type have been studied as factors that may explain the epidemic of overweight and obesity in children. Nowadays, less than 10% of children meet the guideline of 60 min of physical activity per day [13,14]. To avoid long-term persistence of this behavior, which can result in the development of overweight/obesity in children, the majority of intervention programs include as a major target the increase in physical activity [15].

Finally, most of the trials investigating the effectiveness of family interventions, are carried out in clinical, community, or school settings, and, consequently, there is a lack of evidence on home-based interventions [8,16]. To address this issue, we developed the “4 your family” program, a family-based intervention including overweight or obese children that targeted the home environment. The main objective was to evaluate the effectiveness of three different family-based interventions, aimed to decrease children’s body mass index (BMI) z-score, total body fat (TBF) percentage, and waist circumference (WC) z-score. Secondarily, the study aimed to explore differences in factors associated with childhood overweight and obesity, including dietary intake, level of physical activity, and total screen time.

## 2. Materials and Methods

### 2.1. Study Design

The “4 your family” study took place from September 2014 to September 2015. Children aged 8–12 years, who were overweight or obese, as well as their parents, were eligible for the study. Recruitment process included communication at all public schools of Athens as well as through the Unit of Childhood Obesity of the First Department of Pediatrics, National & Kapodistrian University of Athens, “Aghia Sophia” Children’s Hospital. In addition, advertisement of the intervention study was provided through the internet and television. More specifically, the recruitment process was performed via the use of articles written in school newsletters, emails sent to schools, communication with physicians, cooperation with the First Department of Pediatrics of the National and Kapodistrian University of Athens, at “Aghia Sophia” Children’s Hospital, distribution of flyers, TV appearances in morning shows, participation in radio shows, and via a social media campaign. Healthy weight children, as well as children with genetic or chromosomal disorders and with chronic physical (allergies, dietary restriction) or mental health problems, were excluded from the study. Figure 1 shows the study participant flow chart.

As presented in Figure 1, a total of 115 children enrolled in the study and 91 completed it (79% compliance), with all groups having >23 children (36 in Agricultural University of Athens, 30 Private Diet Offices, 25 Website). All participants (and their parents) were informed about the aims and procedures of the study, and a parental signed consent form was provided prior to study inclusion. Ethical approval was provided by the Ethics Committee of the Department of Food Science and Human Nutrition of the Agricultural University of Athens according to the Declaration of Helsinki guidelines.

The study aimed to promote a sustainable, healthy (diet, physical activity, cooking, well-being) approach to family-wide lifestyle change. Two educational kits were provided to the participants at the beginning of the intervention. The children’s educational kit contained a three-dimensional Nutrition–Physical Activity–Food Safety Pyramid and a booklet with Nutritional–Physical Activity–Behavioral–Healthy Cooking tips for children. The parents’ educational kit contained a booklet with Nutritional–Physical Activity–Behavioral–Healthy Cooking tips for parents and an information leaflet with supplemental information and guidance for the intervention program.

Finally, a pedometer and other sport equipment (balls, gym mattresses etc.) were given to each child in order to monitor their steps per day and encourage an increase in physical activity.

The sample size was based on a power of 80% and a level of significance (α) of 0.05 to detect a BMI z-score difference of 0.3 (±0.49) from baseline to 6 months. This resulted in a total sample of 23 children per group, for a total sample of 83. The dropout rate after randomization was assumed to be 20% (consistent with an average attrition rate of 19.7%) [17,18].

### 2.2. Study Procedure

Study duration was 6 months, as this time interval has been recommended as an adequate duration to acquire adequate intervention, since although the duration of intervention is related to more powerful outcome effects, it is not a significant moderator of treatment effects [17,18]. The content of the sessions-combined approaches targeted the child and the parents with techniques known to be effective from parenting programs and family lifestyle programs (i.e., discussion, role play, goal setting, information on healthy eating, food categories, food labels, trying new foods, practical food preparation, behavior modification—Social Cognitive Theory (SCT)—and focus on physical activity) [19].

Parents were encouraged to acquire authoritative parenting skills, meaning persistent and firm, but at the same time indulgent. They were also encouraged to practice healthy behaviors and improve the self-efficacy of their child’s healthy behaviors [19].

The study included 3 interventions and took place in 3 different settings. Specifically (i) Agricultural University of Athens (AUA) (Group 1, *n* = 36), (ii) private dietary offices (Group 2, *n* = 30), and (iii) at home via a specifically developed Web site (Group 3, *n* = 25). Participants were randomly assigned in one of the three groups upon completing the initial assessment, using a computer-generated random numbers table. Detailed information can be seen in a Appendix A. A small summary per intervention group is depicted below.

Group 1 attended 12 training bi-weekly 1-h sessions delivered by 4 experts: a dietitian, a psychologist, a physical education trainer, and a chef. The sessions were delivered simultaneously, in parallel groups, and various tools, such as brochures, books, games, food models, and multimedia aids (presentations, audio-tape messages) were used. All children of this group attended the sessions together, separated from their parents who also attended group sessions. The sessions’ content was the same for parents and children in order to promote greater understanding and discussion at home.

Group 2 followed the same structure intervention, but the latter was delivered face- to-face by trained dietitians to parent and child, with 12 bi-weekly, 30-min sessions, in one of the 6 cooperative Diet Centers in the area of Athens. These sessions contained the same information but were adapted accordingly and were administered separately to parents and children. Prior to study commencement, study-associated dietitians attended a one-day training course provided by the 4 experts (dietitian, psychologist, physical education trainer, and chef) and received a manual, which covered the content, instructions, methodology, and philosophy of the treatment protocol in order to follow a common dietetic practice. These topics were covered in specific orderly sessions, by all dietitians, and forms were provided for completion by both the dietitian and the parent at the end of each session to address compliance to the protocol.

Group 3 attended the same structure intervention from home, being given a specifically designed web page delivered by the same 4 experts: dietitian, psychologist, physical education trainer, and chef, with 12 bi-weekly videos. Each video and presentation lasted 20 to 30 min. Children and parents of each family attended the sessions together. There was a specific web page, which provided all the information about child and parent browsing and participating at the program web page.

### 2.3. Measurements

Anthropometry was performed in all children during the first visit (baseline) and at the end of the intervention period by trained dietitians. More specifically, body weight and total body fat percentage were measured using a digital scale and a Body Composition Analyzer (Tanita TBF-300A, Tokyo, Japan), with children wearing light clothing and no shoes. Height was measured using a portable stadiometer (Seca, Hamburg, Germany) to the nearest 0.1 cm, with the head positioned in the Frankfort plane. Waist circumference was measured to the nearest 0.1 cm using an ergonomic circumference measuring tape (Seca 201, Hamburg, Germany). Children were categorized as overweight or obese based on UK 1990 BMI charts [20]. BMI z-score and waist circumference z-score were calculated using—Centers od Disease Control (CDC), Percentile Data Files with LMS (Lambda for the skew, Mu for the median, and Sigma for the generalized coefficient of variation) Values, and World Health Organization (WHO) growth references 5–19 years [21,22,23].

### 2.4. Dietary Assessment

A validated semi-quantitative Food Frequency Questionnaire (FFQ) containing 76 foods usually consumed was used to assess dietary habits in Greece [24]. Detailed instructions and examples were given in each case by the dietitians participating in the study, as per protocol. Response options included “never”; 1–4 times per week; once a day; 2–3 times a day; 4 or more times a day. These 76 foods were organized in 10 food groups, including: fruit, vegetables, grains and cereals, dairy (milk and milk products), fish, processed meat, sweets, sugar-sweetened beverages (ssbs), and fast food. The FFQ was given for completion at the beginning of the study (baseline) and at the end, after 6 months. Frequency of intake was transformed into daily intake in both cases, and it was multiplied with the portion sizes provided at the end of the questionnaire for each food to obtain total grams per day for each food group [24]. Food intake was derived for each FFQ, and the differences from baseline of the food groups assessed were calculated. Total energy intake (Kcal/day) at baseline and at the end of the study were also derived.

### 2.5. Physical Activity Assessment and Sociodenographic Variables

All children were asked to complete the Physical Activity Questionnaire (PAQ) [25], which is a validated tool for level of physical activity assessment in children. Briefly, it consists of a structured daily or weekly diary on types of activities performed, and their frequency and duration, to discern moderate through vigorous physical activity (MVPA) in hours per day. Total hours per day spent on screen (watching television/DVD/videos and/or recreational usage of games consoles/computer) during weekdays and weekends was also assessed from this questionnaire, and average screen time was calculated ((total hours in weekdays + total hours in weekends)/7). Participants were asked to complete the PAQ during the first visit and after 6 months at the end of the intervention.

Parents were asked to report the educational level and their occupational status, in order to acquire information on basic sociodemographic variables that could affect the results of an intervention. Educational level was grouped in three categories: ≤6 years (elementary); >6 years and ≤12 years (secondary); and >12 years (vocational or higher education). Vocational compared to higher (university) education were not separately grouped due to restrictions of the sample size. Over grouping can incorporate Type 2 error in the analysis. Occupational status was categorized as employed, unemployed, or homemakers.

### 2.6. Statistical Analysis

Categorical variables are presented as frequencies and percentages (*n*, %) and were assessed using a Pearson chi-squared statistic. All continuous variables were checked for normality using the Kolmogorov–Smirnov or Shapiro–Wilks test before further analyses were conducted. Continuous variables with normal distribution are presented as mean ± Standard Deviation (SD), while skewed variables are presented as median (25th percentile, 75th percentile). Within-group comparisons for differences from baseline were performed using a paired-samples Student T-test or related-samples Wilcoxon signed rank test, respectively. Between-group comparisons were performed using one-wayAnalysis Od Variance (ANOVA) for normally distributed variables and Kruskal–Wallis test for those that were skewed. Where group differences were found, post hoc analysis was performed using Tukey’s test (after ANOVA) or Dunnett’s test (after Kruskal–Wallis) to derive specific between-group differences (which group had the greatest difference).

Data entry and calculations were carried out using SPSS v23 software (SPSS for Windows, SPSS Inc., Chicago, IL, USA), and significance level was set at the 5% level.

## 3. Results

Of the 115 children who started the program, 91 (79%) completed it, 15 (13%) partially completed it (attended half the sessions, but attended irregularly), 5 (4%) had lack of interest, and 4 (4%) dropped out for unknown reasons (Figure 1). Program engagement was better in Group 2, with 88% complete attendance (only 4 children withdrew). In Group 3, a total of 10 children withdrew, resulting in 71% attendance, and similarly in Group 1, with 78% complete attendance achieved (10 children withdrew).

Participant baseline anthropometric parameters with respective calculated z-scores are given in Table 1. In addition, baseline calculated dietary energy intake, and physical activity, screen time, as well as parental educational level and occupational status are also tabulated, in all cases by intervention group. No statistically significant differences were observed in any baseline variable among the three intervention groups.

Changes in children’s adiposity, anthropometric parameters, energy, physical activity, and screen time for the three intervention groups over the course of the study are shown in Table 2 and Figure 2. Within-group comparisons indicated significant improvement in body weight, BMI z-score, and screen time from baseline in all three study groups, waist circumference z-score mostly in Group 2, and borderline in Group 3 (gray zone *p* = 0.045), whereas %TBF improved only in Groups 2 and 3.

Between-group differences varied. In particular, %TBF decrease was larger in Group 3, but this decrease was clinically marginal, as it was only 2.1% from baseline compared to 1.9% in Group 2. BMI z-score and screen time changes were significantly larger in Group 2 than in the other two Groups. In addition, children in Group 3 were the only ones who significantly decreased their total dietary energy intake. From the results shown in Table 2, it can be suggested the most pronounced improvements in anthropometric indices were observed in Groups 2 and 3 and not in Group 1.

Dietary intakes of children at the beginning and at the end of each study period are given in Table 3. Within-group differences indicated a slight improvement in vegetable consumption in Group 1 (gray zone; *p* = 0.048), significant statistical increases in whole wheat cereals and grains, low-fat dairy, and decreases in sweets, fast food, and processed meat intakes in Group 2, and increases in fruit and low-fat dairy intakes in Group 3. Between-group comparisons indicated differences in fruits and low-fat dairy, sweets, and processed meat intakes. Finally, post hoc analysis showed that fruit intake improved the most in Group 1, while Group 2 had the highest increase in low-fat dairy selection and the largest decrease in sweets and processed meat intakes. As a whole, it could be suggested that more improvements were observed in Group 2.

## 4. Discussion

The “4 your family” intervention study showed that family-based interventions can result in favorable outcomes not only in regard to body weight and fat status, but also in dietary and behavioral habits, with the most effective approach being overall the one that is personalized. In summary, all interventions were successful, and although between-group differences were found in various outcomes, with Group 2, the intensively individualized group, had the most effective results, while the web-based intervention followed. The parent and child group intervention plan showed the smallest effect in all aspects. This may be explained by two factors: time restrictions parents may have for maximum engagement, and general healthy guidelines provided to all children irrespectively of their underlying problem(s). This outcome suggests that public health intervention programs should be personalized to a population’s needs and characteristics. Specifically, although basic lifestyle factors, such as screen time and physical activity, are easily understood and altered, specific dietary habits require greater planning. It may be beneficial to screen and classify children and parents, by mutual characteristics, nutritional habits, behaviors, and preferences, altering the intervention plan accordingly. This will personalize the intervention to specific family requirements. It should be underlined that there were no significant differences in parental educational level between groups, nor was one found in occupational status. Therefore, this could not have been a potential reason influencing the study’s results.

Family-based interventions have shown important effects immediately at the end of the intervention according to recent reviews [11,26,27]. Overall, it is accepted that family and home environments shape early health habits, and parents play a crucial role in developing their children’s nutrition and physical activity behaviors; parents determine the types of foods available in their homes and provide opportunities to be active [15], so these environments are very important in the aetiology and prevention of childhood obesity. In addition, according to a recent systematic review, including parents in interventions with nutrition and physical activity education sessions could result in a successful improvement of their children’s lifestyle [28] as per this study’s findings.

Many studies have also shown a decrease in BMI z-score [29,30]. Similarly, the most recent reviews that summarize the results of 37 trials in 4019 children showed that the BMI z-score on average was 0.06 units lower in the intervention groups than the control groups [11,26]. In our study, the largest reduction in BMI z-score was observed in Group 2, underlining the need to personalize a nutritional and behavioral intervention. However, although it could be argued that the improvements are small, it has to be emphasized that even a modest reduction in BMI z-score (>0.00–<0.1) is expected to have clinical benefits, and it could be associated with significant improvement in various cardiovascular risk factors [31].

The decrease in TBF% was significant only in the groups that had an “individualized” plan. Specifically, a decrease in TBF% was observed in Group 2, where dietitians were giving individualized advice and in Group 3, where children were taking the initiative to learn any relevant information themselves from a specific link in the web. This can be explained by the fact that these groups not only decreased their time on a screen and increased their physical activity but also significantly improved their dietary habits. In particular, Group 3 decreased energy intake and increased fruit and low-fat dairy intakes, whereas Group 2 showed the largest multilevel nutritional changes, not only as an increase in healthy whole wheat, fruit, vegetable, and low-fat dairy choices, but also in a decrease in obesogenic foods, such as sweets, fast food, and processed food. Surprisingly though, energy intake was not altered in Group 2. Differences in energy intake can partly explain the slightly higher decrease in TBF in Group 2. Regarding decreases in TBF, another family-based treatment failed to observe a significant difference [29], underlining again the need for more personalized interventions. In addition, in accordance to this study, specific dietary changes in line with recommendations have been found by various studies, including an increased consumption of fruits and vegetables, whole grains [32], low-fat [33,34,35], and a decrease in sweets, fast food [36,37], processed meat [34], and sugar-sweetened beverages [38]. Lastly, the increase in the duration of physical activity and the significant reduction in screen time found in all three intervention groups are quite important, as these differences explain partly the observed amelioration in the anthropometric parameters, which is in agreement with other studies that have shown that changes in physical activity and/or screen time can reduce obesity rates in children [39,40,41].

Although this is a study that was conducted in Greece, it is of global importance, since the increasing trend in the prevalence of overweight and obesity in children continues in the US [42] as well as in most southern European countries [43]. Therefore, the epidemic is not of local but of global interest, and results from this study that include parental engagement as well as personalized intervention can increase the effectiveness of interventions.

The main strength of the study was the nature of its design: a Randomized Clinical Trial (RCT) with a standardized methodology was used in all intervention groups. In addition, the study aimed to evaluate three different types of intervention to assess the most beneficial approach for ameliorating anthropometric measures, lifestyle behavior, and nutritional intake. However, this study has limitations also, including losses to follow up, although the percentage lost had been accounted for at the study design stage when calculating sample size for adequate power. Another potential limitation is the fact that this study enrolled participants based on parental evaluation of their child’s weight status, with most parents being of medium or higher educational level. It has been reported that parents of low educational level tend to underestimate their child’s weight and perceive them as normal weight although they are overweight or obese [44]. Therefore, results should carefully be generalized to medium to high educational level families. Another limitation is the fact that there was no follow up to observe the percent of the population that continued to comply with the modifications and retain the achieved anthropometrics.

## 5. Conclusions

In conclusion, personalized, family-based interventions can successfully improve children’s lifestyle habits, as well as body weight and percent body fat status in the long run. Empowering parents with knowledge and skills is important; considering the time restrictions they may have, web-based interventions or a combination of different types of intervention may be more effective in maximizing their involvement.

## Figures and Tables

**Figure 1 nutrients-13-00341-f001:**
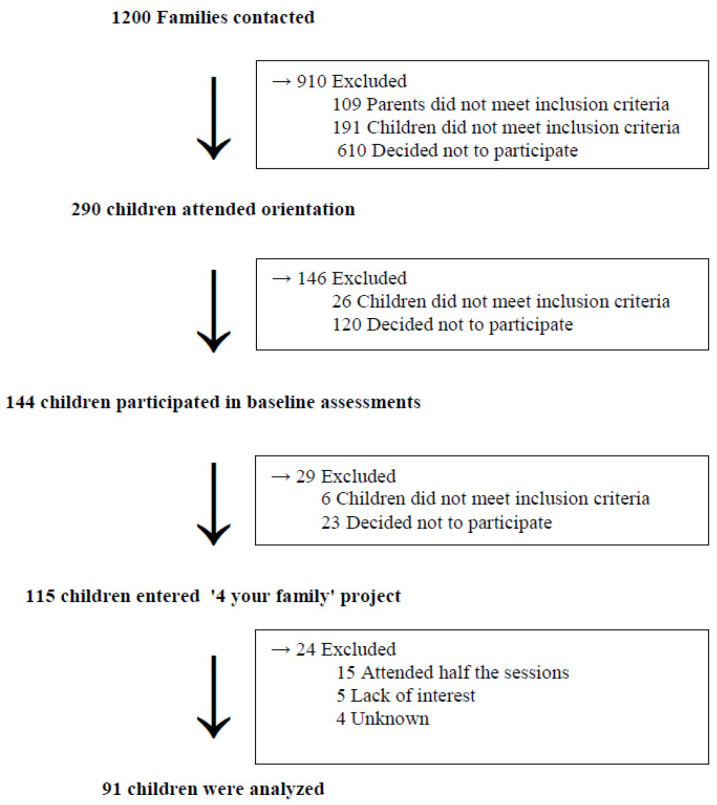
Study participant flow chart.

**Figure 2 nutrients-13-00341-f002:**
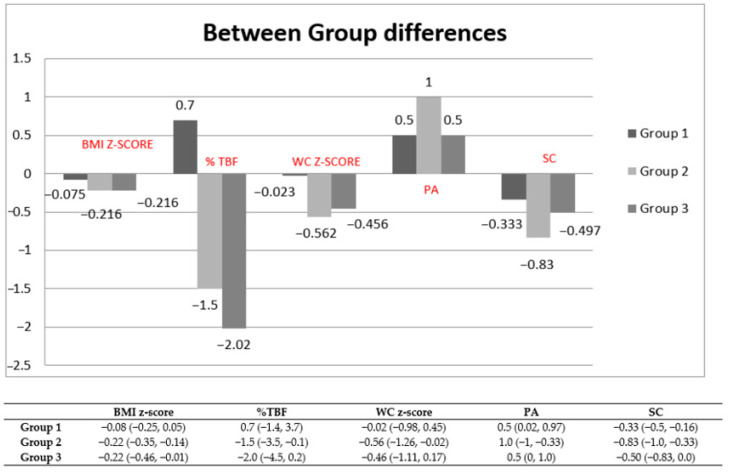
Mean changes of anthropometric, physical activity and screen time measures with between-group differences. Results are presented as median and interquartile range (25th–75th percentile). *p*-values derived through one-way ANOVA test or Kruskal–Wallis H-Test (differences between-group comparisons) and pairwise comparisons (post-hoc tests) Tukey or Dunett Test where appropriate. BMI, body mass index, WC, waist circumference, %TBF, total body fat, PA, physical activity, SC, screen time.

**Table 1 nutrients-13-00341-t001:** Baseline characteristics of the children and demographic parents of parents who participated at the “4 your family” project.

Characteristics	Group 1	Group 2	Group 3	*p*-Value
Children (*n*)	36	30	25	
Sex (Males, *n* (%))	15 (42%)	14 (47%)	10 (40%)	*p* = 0.869
Age (years)	10.0 (9.0, 11.8)	10.0 (9.0, 11.0)	11.0 (9.5, 11.0)	*p* = 0.763
Weight (kg)	55.4 (49.7, 60.6)	56.7 (44.7, 67.1)	54.5 (48.8, 63.7)	*p* = 0.977
Height (m)	1.46 (0.08)	1.46 (0.09)	1.47 (0.09)	*p* = 0.905
Obesity, *n* (%)	23 (64%)	21 (70%)	15 (60%)	*p* = 0.733
BMI	25.5 (23.8, 27.8)	26.4 (23.5, 28.6)	25.2 (22.9, 28.4)	*p* = 0.953
BMI z-score	2.73 (0.6)	2.66 (0.5)	2.6 (0.8)	*p* = 0.722
Total body fat (%)	34.4 (6.6)	34.1 (6.5)	34 (7.1)	*p* = 0.970
Waist circumference (cm)	87.6 (81.6, 91.3)	86.4 (77.5, 91.2)	83.0 (77.5, 91.2)	*p* = 0.342
Waist circumference z-score	5.0 (4.2, 6.2)	5.03 (3.7, 5.5)	4.21 (3.1, 5.1)	*p* = 0.108
Energy (kcal)	1774 (402)	1591 (284)	1761 (388)	*p* = 0.096
Physical activity (h/d)	1.5 (1.0, 1.9)	1.8 (1.5, 1.9)	1.8 (1.3, 2.7)	*p* = 0.077
Screen time (h/d)	2.3 (1.5, 4.0)	3.0 (2.0, 3.6)	2.7 (1.7, 3.0)	*p* = 0.399
Mother’s education (*n* (%))				
≤6 years	0 (0%)	1 (3%)	2 (8%)	
≤12 years	16 (44 %)	7 (24%)	11 (44%)	*p* = 0.145
>12 years	20 (56 %)	22 (73%)	12 (48%)	
Father’s education (*n* (%))				
≤6 years	3 (8%)	2 (7%)	6 (24%)	
≤12 years	16 (45%)	8 (27%)	4 (16%)	*p* = 0.053
>12 years	17 (47%)	20 (66%)	15 (60%)	
Mother’s employment (*n* (%))				
Unempoloyed	6 (17%)	2 (7%)	2 (8%)	
Homemaker	6 (17%)	11 (36%)	9 (36%)	*p* = 0.276
Employed	24 (66%)	17 (57%)	14 (56%)	
Father’s employment (*n* (%))				
Unemployed	3 (8%)	3 (10%)	3 (12%)	
Homemaker	0 (0%)	0 (0%)	0 (0%)	*p* = 0.894
Employed	33 (92%)	27 (90%)	22 (80%)	

One-way Analysis of Variance (ANOVA) test for normal distributed variables, Kruskal–Wallis H Test for skewed variables and Pearson chi-squared tests for categorical variables; Results are presented as mean (SD) or median interquartile range (25th–75th percentile) for the skewed variables. BMI: body mass index; Obesity: based on BMI z-score categorization.

**Table 2 nutrients-13-00341-t002:** Anthropometric parameters, physical activity and screen time at baseline and at the end of the 6-month intervention period by intervention group.

Children (*n*)	Group 1 *n* = 36			Group 2 *n* = 30			Group 3 *n* = 25				
	0 Month	6 Months	*p*-Value ^a^	0 Month	6 Months	*p*-Value ^b^	0 Month	6 Months	*p*-Value ^a^	*p*-Value ^b^	Post-Hoc ***
Weight (kg)	57.1 (11.9)	61.1 (12)	0.000	56.2 (13.1)	57.8 (14.5)	0.026	56.5 (11.5)	58.9 (11.7)	0.000	0.016	Group 3
Height (m)	1.46 (0.08)	1.51 (0.08)	0.000	1.46 (0.09)	1.49 (0.10)	0.000	1.47 (0.09)	1.51 (0.09)	0.000	0.375	
BMI (kg/m^2^)	26.2 (3.4)	26.6 (3.5)	0.362	26.0 (3.8)	25.3 (4.2)	0.007	26.0 (3.9)	25.4 (3.8)	0.069	0.064	
BMI z-score	2.73 (0.6)	2.60 (0.6)	0.009	2.66 (0.5)	2.38 (0.7)	0.000	2.60 (0.8)	2.36 (0.7)	0.000	0.016	Group 2
%TBF	34.4 (6.6)	35.6 (6.6)	0.057	34.1 (6.5)	32.2 (7.8)	0.013	34.0 (7.1)	31.9 (7.8)	0.020	0.001	Group 3
WC (cm)	87.6 (7.5)	87.2 (8.5)	0.831	85.2 (9.7)	82.2 (10.5)	0.000	85.2 (9.8)	83.5 (8.5)	0.161	0.197	
WC z-score	5.27 (1.39)	4.99 (1.51)	0.135	4.70 (1.54)	3.97 (1.71)	0.000	4.63 (1.85)	4.18 (1.71)	0.045	0.112	
Energy (kcal)	1774 (402)	1701 (403)	0.078	1591 (284)	1520 (266)	0.136	1761 (388)	1631 (354)	0.000	0.334	
Physical Activity (h/d)	1.47 (1.0, 1.8)	1.7 (1.5, 2.4)	0.000	1.8 (1.5, 1.9)	2.5 (1.8, 2.8)	0.000	1.8 (1.2, 2.6)	2.2 (1.5, 3.1)	0.000	0.255	
Screen time (h/d)	2.3 (1.5, 4.0)	2.4 (1.5, 3.4)	0.000	3.0 (2.0, 3.6)	2.0 (1.0, 3.1)	0.000	2.7 (1.7, 3.0)	2.0 (1.0, 3.0)	0.001	0.010	Group 2

Results are presented as mean (SD) for the normally distributed variables or as median interquartile range (25th–75th percentile) for the skewed variables. ^a^
*p*-Values derived through paired-samples Student t-test or related-samples Wilcoxon signed-rank test (within-group comparisons). ^b^
*p*-values derived through one-way ANOVA test or Kruskal–Wallis H-test (mean differences tested between-group comparisons). * post-hoc Tukey or Dunnett test: Group with greatest difference. BMI, body mass index. WC, waist circumference. %TBF, percent Total Body Fat.

**Table 3 nutrients-13-00341-t003:** Summary of food group intake in the beginning at the end of the intervention period in servings per day.

Food Category (Servings per Day)	Group 1			Group 2			Group 3				
	0 Month	6 Months	*p*-Value ^a^	0 Month	6 Months	*p*-Value ^a^	0 Month	6 Months	*p*-Value ^a^	*p*-Value ^b^	Post Hoc *
Fruit	1.00 (0.43, 2.00)	1.00 (0.42, 2.00)	0.944	1.00 (0.43, 2.00)	1.00 (1.00, 2.00)	0.091	0.43 (0.21, 1.75)	1.00 (1.0, 2.75)	0.000	0.001	Group 3
Vegetable	1.00 (0.42, 2.00)	1.00 (0.43, 1.75)	0.048	0.71 (0.11, 1.00)	1.00 (0.42, 2.00)	0.003	1.00 (0.21, 1.00)	1.00 (0.42, 1.75)	0.219	0.448	
Whole wheat cereals/grains	0.51 (0.14, 1.62)	0.65 (0.17, 1.24)	0.673	0.44 (0.13, 0.93)	0.76 (0.25, 1.28)	0.000	0.19 (0, 0.83)	0.19 (0, 1.00)	0.266	0.077	
Dairy, low fat	1.06 (0.24, 2.00)	1.14 (0.69, 1.56)	0.866	1.00 (0.35, 1.45)	1.29 (0.89, 2.00)	0.001	0.83 (0.40, 1.21)	1.00 (0.35, 1.85)	0.034	0.016	Group 2
Legumes	0.14 (0.14, 0.28)	0.14 (0.07, 0.28)	0.567	0.14 (0.05, 0.28)	0.14 (0.14, 0.29)	0.338	0.14 (0.14, 0.28)	0.14 (0.05, 0.30)	0.500	0.328	
Fish	0.14 (0.05, 0.18)	0.14 (0.05, 0.14)	0.120	0.14 (0.05, 0.14)	0.14 (0.13, 0.14)	0.168	0.14 (0.01, 0.14)	0.10 (0.05, 0.28)	0.527	0.075	
Sweet	0.47 (0.24, 0.86)	0.57 (0.30, 0.80)	0.545	0.49 (0.29, 0.99)	0.40 (0.29, 0.85)	0.011	0.62 (0.26, 0.86)	0.80 (0.61, 1.12)	0.074	0.026	Group 2
Fast food	0.14 (0.05, 0.23)	0.19 (0.10, 0.23)	0.282	0.14 (0.10, 0.19)	0.10 (0.05, 0.19)	0.004	0.10 (0.05, 0.31)	0.15 (0.10, 0.26)	0.695	0.018	Group 2
Processed meat	0.05 (0, 0.28)	0.10 (0.01, 0.37)	0.807	0.1 (0.025, 0.64)	0.05 (0, 0.35)	0.026	0.20 (0.02, 0.33)	0.19 (0.02,0.46)	0.527	0.095	
Sugar-sweetened beverages	0.05 (0, 0.14)	0.05 (0, 0.14)	0.277	0.05 (0, 0.14)	0.00 (0, 0.05)	0.160	0.05 (0, 0.05)	0.05 (0, 0.09)	0.775	0.710	

Results are presented as median and interquartile range (25th–75th percentile) for the skewed variables. ^a^
*p*-values derived through related-samples Wilcoxon signed-rank test (within-group comparisons).^b^
*p*-values derived through Kruskal–Wallis H-Test (differences between-group comparisons). * post hoc Tukey or Dunnett test: Group with greatest difference. Dairy low-fat: milk, yoghurt, cheese, Processed meat: chicken ham, pork ham, salami, Fish: tuna, sardine, octopus, calamari, Legumes: beans, fava, lentils, Whole wheat cereals/grain: cereals, bread, pasta, rice, Sweet: all kinds, Fast food: burger, pizza, souvlaki, pie with cheese or vegetables.

## Data Availability

Raw data were generated at (Agricultural University of Athens). Derived data supporting the findings of this study are available from the corresponding author (A.Z.) on request.

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
