# Peer review of "Effects of Three Different Family-Based Interventions in Overweight and Obese Children: The “4 Your Family” Randomized Controlled Trial"

_nutrients, 2021, doi:10.3390/nu13020341_

Round 1
Reviewer 1 Report
The current study signifies the importance of engaging family in interventions targeted for children with unhealthy weight. The study design is robust and 'Results' looks promising. There are few key elements missing in 'Methods' & 'Discussion' section which are detailed below;
Methods
1. Since the Readers are from an international context, a little more geographical context here would be informative.
2. Demographic factors of parents are missing. Basic social and economic characteristics of the families that participated is important information as those factors interplay with their lifestyle practices and decisions.
3. Each intervention type requires more detailed description.
For example; it is stated that Group 1 attends sessions separately for Parents and Children; was the content delivered to both groups the same? Who delivered it to the children cohort? What were the resources used? Was the resources developed specifically for this study? Group 2 individual sessions were together or separate for parent and child?
4. Line 114: Is this statement entirely true? There is evidence that indicate, longer duration of interventions (more than 6 months) might be a good indicator to measure sustained intervention outcomes.
Results
5. Figure 1 - at present text boxes hide the plain text below. Please edit the Figure.
Discussion
6. It is important to add the Social and environmental factors that the families habituate. Please add the relevance of 'social determinants of health and lifestyle practices' into the Discussion.
7. Line 325: Agree the study has strengths. However, it is good to reflect on study limitations for readers to understand the gaps and for future research purposes. Small study duration, lack of information regarding Family demographics etc. are few examples of limitations observed.
Author Response
Reviewer 1
Comments and Suggestions for Authors
The current study signifies the importance of engaging family in interventions targeted for children with unhealthy weight. The study design is robust and 'Results' looks promising. There are few key elements missing in 'Methods' & 'Discussion' section which are detailed below;
Authors: Thank you very much for your positive feedback and valuable comments. The authors have reviewed all points raise and have tried to respond and improve the manuscript accordingly.
Methods
- Since the Readers are from an international context, a little more geographical context here would be informative.
Authors: Thank you for the comment. The authors have added the following in the discussion section. “Although this is a study that was conducted in Greece, it is of global importance since the increasing trend in prevalence of overweight and obesity in children continues in the US [42] as well as in most southern European countries [43]. The epidemic is therefore not of local but of global interest, and results from this study that include parental engagement as well as personalized intervention can increase effectiveness of interventions.”
- Demographic factors of parents are missing. Basic social and economic characteristics of the families that participated is important information as those factors interplay with their lifestyle practices and decisions.
Authors: Thank you very much for your comment. The information for educational level and occupational status have been in Table 1 (baseline characteristics…) added since, as the Reviewer correctly noted, this could have acted as a confounding factor. Results however depict that there were no significant differences in educational level between groups in maternal nor paternal educational level, nor was one found in occupational status. Therefore, this could not have been a potential reason influencing the study’s results. This comment has also been added in the results and the discussion section.
- Each intervention type requires more detailed description.
For example; it is stated that Group 1 attends sessions separately for Parents and Children; was the content delivered to both groups the same? Who delivered it to the children cohort? What were the resources used? Was the resources developed specifically for this study? Group 2 individual sessions were together or separate for parent and child?
Authors: A supplementary table with a detailed description for each session and for each group of intervention has been constructed and uploaded depicting in detail the process and the resources used.
- Line 114: Is this statement entirely true? There is evidence that indicate, longer duration of interventions (more than 6 months) might be a good indicator to measure sustained intervention outcomes.
Authors: We thank the Reviewer for the comment. The sentence was modified grammatically to avoid misconceptions that the previous phrase may have led to. It now states: “Study duration was 6 months, as this time interval has been recommended as an adequate duration to acquire adequate intervention, since although duration of intervention is related to more powerful outcome effects, it is not a significant moderator of treatment effects.”
Results
- Figure 1 - at present text boxes hide the plain text below. Please edit the Figure.
Authors: The figure has now been edited. Apologies.
Discussion
- It is important to add the Social and environmental factors that the families habituate. Please add the relevance of 'social determinants of health and lifestyle practices' into the Discussion.
Authors: Thank you for the comment. This has now been added.
- Line 325: Agree the study has strengths. However, it is good to reflect on study limitations for readers to understand the gaps and for future research purposes. Small study duration, lack of information regarding Family demographics etc. are few examples of limitations observed.
Authors: Thank you for the comment. Some limitations had been written in the original submission although they were not clear. These have now been clarified and the edited (lines 430-452).
Reviewer 2 Report
Dear Authors,
The manuscript (nutrients-1085958) presented for review is very interesting and I recommend the article for publication after major revision.
Authors, Please note and address the following comments:
Abstract:
Lines 30-32: The following text is unnecessary.
Keywords: keyword 1; keyword 2; keyword 3 (List three to ten pertinent keywords specific to the article yet reasonably common within the subject discipline).
Introduction:
In my opinion, the introduction does not demonstrate a gap in the literature. The Introduction and discussion chapters are the weakest parts of the work.
Has anyone before the authors dealt with home dietetics interventions?
Have any authors earlier conducted an online dietetics family-based intervention?
What were the results of other family-based interventions in Greece?
Results
Figure 1 (page 5) is unclear. It should be corrected.
Figure 2 (page 8) – The abbreviation descriptions below Figure 2 is missing.
Does the table below Figure 2 complete it?
Maybe it is better to transfer the data from a Table to a Figure or just use only a Table?
I don't understand what the authors present in brackets in the Table? Standard Deviation?
For example:
|
|
BMI z-score |
|
Group 1 Group 2 Group 3 |
-0.075 (-0.25, 0.05) -0.216 (-0.35, -0.14) -0.216 (-0.46, -0.01) |
Lines 245-247: It does not sounds scientific: “In particular, %TBF decrease was larger in Group 3, but this decrease was clinically marginal as it was only 2.1% from baseline compared to 1.9% in Group 2…..”
Were these differences statistically significant or not?
Table 3 (page 8) – the same situation as in Table (without number) below Figure 2. I don't understand what the authors present in brackets in the Table? Standard Deviation?
Discussion
The Discussion section isn't made clear. I miss a summary. In the discussion, the authors mainly emphasize the differences between the different family groups but do not try to explain why such differences occur. Without knowing the education and financial status of the parents, it is difficult to conclude what is the reason for not following the recommendations. Perhaps low education of parents, or maybe just lack of money.
Conclusion
What are the practical and theoretical implications of the research?
There is no answer to what extent this online intervention was effective. The current conclusions are quite enigmatic.
Literature
The citations of literature in this manuscript are in a top index, but it should be in square brackets and numbered in order of appearance in the text.
As the following rules in Nutrients Journal:
“In the text, reference numbers should be placed in square brackets [ ] and placed before the punctuation; for example [1], [1–3] or [1,3]. References must be numbered in order of appearance in the text (including citations in tables and legends) and listed individually at the end of the manuscript”.
References are not saved properly. It is not according to the Nutrients journal rules.
Despite my comments, I am pleased to recommend this manuscript for publication. I believe it addresses an important area of research in an international context.
Reviewer
Reviewer
Author Response
Reviewer 2
Dear Authors,
The manuscript (nutrients-1085958) presented for review is very interesting and I recommend the article for publication after major revision.
Authors: We thank the Reviewer for his/her comments and for the recommendation.
Abstract:
Lines 30-32: The following text is unnecessary.
Authors: based on the article we have on hand, the numbers listed are for the last sentence of the abstract and the keywords. Overall, we believe that the conclusion in the abstract is necessary, and we would be grateful if the reviewer let us keep it.
Keywords: keyword 1; keyword 2; keyword 3 (List three to ten pertinent keywords specific to the article yet reasonably common within the subject discipline).
Authors: We are unclear on the Reviewer’s comment. We though believe that the 3 keywords used are absolutely pertinent to the article and the subject discipline.
Introduction:
In my opinion, the introduction does not demonstrate a gap in the literature. The Introduction and discussion chapters are the weakest parts of the work.
Authors: The Authors have reviewed and edited the Introduction to clearly demonstrate the gap. The discussion has also been modified accordingly.
Has anyone before the authors dealt with home dietetics interventions? Have any authors earlier conducted an online dietetics family-based intervention?
Authors: Two of the authors have primary experience, for over than 20 years, with personalized dietetic interventions and we believe they are adequately qualified to understand whether the process/intervention was effectively conveyed, and the problems that arise respectively.
What were the results of other family-based interventions in Greece?
Authors: To our knowledge other family based interventions that can be compared to this three group approach/methodology has not been implemented in Greece.
Results
Figure 1 (page 5) is unclear. It should be corrected.
Authors: Thank you for your comment. This has been corrected.
Figure 2 (page 8) – The abbreviation descriptions below Figure 2 is missing.
Does the table below Figure 2 complete it?
Maybe it is better to transfer the data from a Table to a Figure or just use only a Table?
I don't understand what the authors present in brackets in the Table? Standard Deviation?
For example:
|
|
BMI z-score |
|
Group 1 Group 2 Group 3 |
-0.075 (-0.25, 0.05) -0.216 (-0.35, -0.14) -0.216 (-0.46, -0.01) |
Authors. The figure has now been edited and a clear description has been provided.
Lines 245-247: It does not sounds scientific: “In particular, %TBF decrease was larger in Group 3, but this decrease was clinically marginal as it was only 2.1% from baseline compared to 1.9% in Group 2…..”
Were these differences statistically significant or not?
Authors: We used the term “statistically significant” as a mathematical term. On the other hand when we used the term “clinical significance”, we related it to the actual health effect this difference makes. The authors clarify that a statistical significant difference of 0.2% in TBF is not of clinical significance and feel that it is more than scientific to state and clarify.
Table 3 (page 8) – the same situation as in Table (without number) below Figure 2. I don't understand what the authors present in brackets in the Table? Standard Deviation?
Authors: Apologies, this is now clarified (interquartile ranges).
Discussion
The Discussion section isn't made clear. I miss a summary. In the discussion, the authors mainly emphasize the differences between the different family groups but do not try to explain why such differences occur. Without knowing the education and financial status of the parents, it is difficult to conclude what is the reason for not following the recommendations. Perhaps low education of parents, or maybe just lack of money.
Authors: a summary is provided in the 1st paragraph. The authors also stated that although all 3 interventions had an effect, the dietitian’s group was most effective overall most probably due to the personalized approach. This has now been entered in the first paragraph as well.
Conclusion
What are the practical and theoretical implications of the research?
Authors: That a more personalized approach should be used when organizing group interventions. This was stated in the primary manuscript.
There is no answer to what extent this online intervention was effective. The current conclusions are quite enigmatic.
Authors: The Authors believe that the conclusions are given clearly suggest personalized nutrition based on common dietary and behavioral characteristics, although all intervention types used had a small effect when they strive for nutritional education and awareness. However, we have edited the conclusion for greater clarification.
Literature
The citations of literature in this manuscript are in a top index, but it should be in square brackets and numbered in order of appearance in the text. As the following rules in Nutrients Journal: “In the text, reference numbers should be placed in square brackets [ ] and placed before the punctuation; for example [1], [1–3] or [1,3]. References must be numbered in order of appearance in the text (including citations in tables and legends) and listed individually at the end of the manuscript”. References are not saved properly. It is not according to the Nutrients journal rules.
Authors: these have been modified as per the Journal’s recommendations.
Despite my comments, I am pleased to recommend this manuscript for publication. I believe it addresses an important area of research in an international context.
Authors: Thank you very much.
Round 2
Reviewer 2 Report
Dear Authors,
The authors have changed many parts of the planned paper according to my suggestions.
I would like to thank the authors for considering my comments and applaud them for the major revisions to improve their manuscript.
But there are still incomprehensible parts of the manuscript as follows:
Lines 197-201: „Parents were asked to report the educational level and their occupational status, in
order to acquire information on basic sociodemographic variables that could affect the results of an intervention. Educational level was grouped into three categories: ≤6 years; >6 years & ≤12 years; and >12 years. Occupational status was categorized as employed, unemployed or homemakers”.
What Authors mean write education <6 years; >6 years ≤12 years; and >12 years. Usually, it is used: elementary school, vocational school, secondary school, higher education (university).
Table 1 For example: Age (years) 10.0 (9.0, 11.75).
I still don't understand what the authors present in brackets in the Table? Is this range?
If it is yes, it should be 10.00 (9-11.75).
The same situation is in Table 2 and Figure 2.
Table 2. For example Physical Activity (h/d) 1.47 (1.0, 1.8)
Figure 2 – For example Group 1 -0.075 (-0.25, 0.05)
The manuscript is interesting and valuable. I recommend the manuscript to be revised after correction of the mentioned above comments.
Reviewer
Author Response
Thank you for your second feedback. The Authors clarify below all points that have not been adequately clarified.
1. Educational level: ≤6 years (elementary school); ≥6 years & <12 (secondary level); ≥12 years (higher education or vocational schooling). Vocational compared to higher (university) education were not separately grouped due to restrictions of the sample size. Over grouping can incorporate Type 2 error in the analysis.
This has been entered into the manuscript.
2. Tables & graphs: the brackets include interquartile range (25% - 75%). This has been clarified in all Tables and graph 2. Decimal points were selected and corrected, according to a logical meaning for a variable presented (ie physical activity to 1 decimal, not 2), unless otherwise necessary. However, unfortunately, the Authors believe these values cannot be separated with "-", since, in case of negative changes, this would raise more questions and consistency needs to be maintained throughout; space (character) however, was included for greater clarity. the decimal points have been chosen and corrected for consistency. Thank you.